# Effects of Communication Skills and Organisational Communication Satisfaction on Self-Efficacy for Handoffs among Nurses in South Korea

**DOI:** 10.3390/healthcare11243125

**Published:** 2023-12-08

**Authors:** Yongmi Lee, Hyekyoung Kim, Younjae Oh

**Affiliations:** 1College of Nursing, Kangwon National University, 1 Kangwondaehak-gil, Chuncheon-si 24341, Gangwon-do, Republic of Korea; rena@kangwon.ac.kr; 2Graduate School of Health Science, Hallym University, 1 Hallymdaehak-gil, Chuncheon-si 24252, Gangwon-do, Republic of Korea; 93660315@hanmail.net; 3Hanil General Hospital, 308 Uicheon-ro, Seoul 01450, Republic of Korea; 4College of Nursing, Research Institute of Nursing Science, Hallym University, 1 Hallymdaehak-gil, Chuncheon-si 24252, Gangwon-do, Republic of Korea

**Keywords:** handoffs, self-efficacy, communication competence, organisational communication, satisfaction, nurse

## Abstract

(1) Background: Although numerous studies related to communication in a nursing context have been conducted, there is a lack of research considering the effects of personal and organisational communication factors on the self-efficacy for handoffs. This study aimed to identify the impact of communication competence and intra-organisational communication satisfaction on self-efficacy for handoffs among nurses. (2) Methods: This cross-sectional research was conducted between September and October 2018. In total, 203 registered nurses were invited to participate in the study by convenience sampling from five general hospitals in South Korea. Data were analysed using SPSS for *t*-test, one-way analysis of variance, Pearson’s correlation coefficients, and multiple regression analysis. (3) Results: In the final regression model, the adjusted R square was significant, explaining 24.2% of the variance in self-efficacy for handoffs (F = 22.43, *p* = 0.001) when the variable horizontal communication (*β* = 0.282, *p* < 0.001) was included in intra-organisational communication satisfaction. In addition, the longer the nurse’s experience in the current unit and the higher the communication competence, the more statistically significant the self-efficacy for handoffs was found to be (*β* = 0.215, *p* = 0.001 and *β* = 0.180, *p* = 0.008). (4) Conclusions: To enhance the self-efficacy for handoffs, nurse managers should foster an atmosphere that allows their staff nurses to interact freely and establish specific guidelines for handoffs through mutual communication.

## 1. Introduction

In light of the escalating significance of patient safety, there is a growing interest within the medical community regarding the clinical handoff processes among healthcare professionals [1]. The World Health Organization [2] and the Joint Commission [3] strongly recommend processes for structured clinical communication handoffs, which various international organisations also support. The nursing handoff is essential for each shift nurse given the nature of shift work for nurses in the ward who care for in-patients continuously. The handoffs between nurses are a specific communication that transfers information, responsibilities, and authority essential to patient care to maintain patient information quality, safety, and continuity [1]. Poor handoffs between nurses, like irrelevant or insufficient handoffs, result in conflicts between nurses and even impact patient care, such as reduced time for delivering nursing care due to extended handoff hours [1,4]. Given that even incorrect handoffs can destroy appropriate continuity of care and increase the risk of nursing error, accurate inter-communication between nurses is critical. Specifically, the failure of nurses to accurately and promptly share all pertinent clinical information about the patient could lead to serious incidents, insufficient care, and delayed treatment. Correct and efficient handoffs are pivotal; thus, self-efficacy for handoffs is being given more emphasis in nursing education [5,6].

Nurses’ self-efficacy for handoffs can be defined as a personal belief in their abilities to effectively transfer patient information and responsibility from one nurse or healthcare provider to another during shift changes or other transitions in care [5]. In the nursing literature, the level of self-efficacy has often been used to measure the effects of an education programme on promoting handoff competence [6,7]. Handoffs are critical for maintaining patient safety, as they ensure continuity of care. The effects of high self-efficacy for handoffs among nurses are multifaceted: improved patient safety [8,9], higher quality of care [8,10], or enhanced teamwork and collaboration [11,12]. Regarding nurses’ well-being, promoted self-efficacy for handoffs can alleviate psychological discomfort related to their tasks, increasing job satisfaction [13,14]. Hence, investigating factors influencing nurses’ self-efficacy for handoffs is crucial.

According to recent reviews on nurse-to-nurse handoffs, it is essential to consider nurses’ communication competence in effective and efficient nursing handoffs [1,7]. Communication competence is one of the core concepts in studies on interpersonal interactions and refers to understanding suitable communication methods for specific contexts and the ability to apply this understanding [1,7]. Effective communication encompasses sharing information; asking for, providing, and confirming details; and addressing emotional and social aspects of communication [15]. Individually, nurses can improve their handoff communication skills by recognising factors that impact the effectiveness of handoffs, irrespective of how an organisation formalises the interaction. Enhancing nurses’ communication skills can result in better handoffs and fewer mistakes related to miscommunication in patient care [8,9]. Indeed, a lack of communication competence can increase anxiety or time stress in handoffs, and reduce the effects and efficiency of handoffs or cause conflicts between nurses [16,17]. In response, organisational or systemic approaches can manage and resolve such individual limitations by establishing guidelines or protocols, encouraging education or training, or improving intra-organisational communication [18,19].

Intra-organisational communication refers to the exchange of information, ideas, and messages among an organisation’s members [20]. This can encompass communication between different departments, teams, or individuals at various levels of the organisation [20]. It can be facilitated through various channels, such as face-to-face meetings, emails, memos, reports, and electronic messaging systems [21]. As such, the quality of intra-organisational communication as perceived by nurses and their satisfaction with it can play an important role in influencing organisational commitment [22] or job satisfaction [23], which can impact the quality of nursing and patient safety [18]. In particular, nurses’ satisfaction with intra-organisational communication comprises four sub-domains (vertical and horizontal communication, media quality, and organisational climate) [5], which have been reported as significant variables in nursing handoffs. For instance, a significant association between the perception of organisational climate among nurses and handoff quality was found [18]; in addition, gaps in personal preferences, one of the causes of ineffective horizontal communication, were reported as barriers in nurse-to-nurse handoffs [19].

Given the above statements, previous studies have indicated that personal communication competence and satisfaction with intra-organisation communication among nurses are essential in nurse-to-nurse handoffs, which can impact good nursing care and patient safety. In this regard, more emphasis should be placed on nurses’ self-efficacy for handoffs. Furthermore, self-efficacy for handoffs is needed to verify these relationships in Korean nurses as they may work in environments that prioritise collectivism or have a hierarchical structure [24,25]. Thus, our study aimed to identify the association between self-efficacy for handoffs and communication competence and between self-efficacy for handoffs and intra-organisational communication satisfaction and the impact of competence and intra-organisational communication satisfaction on self-efficacy for handoffs among nurses.

## 2. Materials and Methods

### 2.1. Participants and Settings

This study was structured using a cross-sectional design to meet its objectives. It involved a convenience sample of 206 full-time registered nurses from five large general hospitals, each with over 200 beds, located in three key regions of South Korea: Seoul, Gyeonggi, and Gangwon Province. Despite the use of convenience sampling, these hospitals comprised various departments, including general, critical, and some specialised units. The sample size was calculated to ensure a significance level of α = 0.05, with 80% power and a medium effect size of 0.15. This calculation was facilitated by the G*Power Program (version 3.1, IBM Korea, Seoul, Republic of Korea). Considering a 20% allowance for potential dropouts and errors, the necessary total sample size was determined to be 206.

Data collection occurred from 20 September to 9 October 2018. The participating nurses were given questionnaires, which they were instructed to seal in envelopes to maintain confidentiality. These were then collected by the researcher either during hospital visits or through mail. Out of the 206 distributed questionnaires, 204 were returned, marking a response rate of 99.0%. After discarding questionnaires with unclear or missing responses, 203 were deemed suitable for the analysis.

### 2.2. Instruments

#### 2.2.1. General Characteristics

The survey questions designed to assess the general characteristics of participants and aspects related to nurse-to-nurse handoffs drew upon previous research [1,7]. These general characteristics were divided into ten categories: gender, age, marital status, religion, level of education, workplace, total years of nursing experience, and duration of employment in the current unit, and, as some of the factors were associated with handoffs, the time allocated for preparing a handoff and the actual handoff duration were also considered.

#### 2.2.2. Self-Efficacy for Handoffs

A Korean version of the Perceived Self-efficacy of Handoff reporting (K-PSH) was employed, which was validated by Hwang et al. [5] for nursing students from the PSH developed by Lee et al. [26]. The authors of the Korean version granted permission to use K-PSH. Given that K-PSH was originally developed for nursing students, a committee of experts was formed to verify the content validity for nurses and it was subsequently supplemented. The expert committee consisted of three university professors majoring in nursing and two primary nurses from general hospitals with more than 25 years of nursing experience. The validated revised version of K-PSH comprises nine items rated on a 6-point Likert scale ranging from 1 (strongly disagree) to 6 (strongly agree) (Appendix A). A higher score indicates a greater level of self-efficacy for handoffs. Lee et al. [26] reported a PSH reliability of 0.86, and [5] found a reliability of 0.93. This study supported the K-PSH reliability with a Cronbach’s alpha of 0.83.

#### 2.2.3. Communication Competence

Global Interpersonal Communication Competence (GICC) was employed, which was revised and validated by Hur [27] from the ICC developed by Rubin and Martin [28]. The authors of the Korean version granted permission to use GICC. GICC is a 15-item scale, and each item is on a 5-point Likert scale ranging from 1 (not at all) to 5 (very high). A higher score indicates greater communication competence. Hur [27] reported a Cronbach’s alpha reliability of 0.72. This study supported the GICC reliability with a Cronbach’s alpha of 0.84.

#### 2.2.4. Intra-Organisational Communication Satisfaction

This study utilised the Korean adaptation of the Intra-Organisational Communication Satisfaction (K-IOCS) scale. The IOCS was initially developed by Downs and Hazen [20], and later, the K-IOCS was validated by Hong [29]. We obtained permission to use the K-IOCS from the Korean authors. This scale comprises 24 items divided into categories like vertical communication, horizontal communication, communication media, and communication climate. Participants rated each item using a 5-point Likert scale, where a score of one point signifies ‘not at all’ and a score of five points indicates ‘very high’. Higher scores reflect greater communication satisfaction. Hong’s [29] validation of the K-IOCS revealed a Cronbach’s alpha of 0.88, while this study found a Cronbach’s alpha of 0.84, confirming the reliability of the K-IOCS.

### 2.3. Data Analysis

This study utilised Cronbach’s alpha reliability analysis, *t*-test, one-way analysis of variance (ANOVA) with post hoc Scheffé tests, Pearson’s correlation coefficients, and stepwise multiple regression analysis. These analyses were performed using SPSS for Windows v. 21.0 (IBM, New York, NY, USA).

### 2.4. Ethical Considerations

This study was conducted following the Declaration of Helsinki and was approved by the Institutional Review Board of Hallym University before data collection (HIRB NO. 2018-039). Written informed consent about the study purpose, including guaranteed anonymity and confidentiality, was obtained; only the participants who voluntarily agreed to participate were sampled, and they were allowed to withdraw at any time without repercussions.

## 3. Results

### 3.1. Differences in Self-Efficacy for Handoffs Based on General Characteristics

Of all the participants, 45.3% were between 26 and 30 years old, with a mean age of 28.5 (SD = 5.6). One hundred and ninety-eight respondents were female (97.5%); over half had no religion (54.2%) and had bachelor’s or graduate degrees (75.4%). The mean of nursing experience was 4.8 years (SD = 5.7), and the mean of work experience in the current unit was 2.3 years (SD = 1.9). The differences in self-efficacy for handoffs varied depending on age, marital status, educational level, years of nursing experience, and years of work in the current unit (*p* < 0.01). Post hoc Scheffé tests were conducted with levels of self-efficacy for handoffs as the dependent variable and age, educational level, years of nursing experience, and years of work experience in the current unit. The results indicated that nurses who were 20–25 years old had significantly lower self-efficacy for handoffs (F = 13.37, *p* < 0.01). Conversely, nurses with a graduate degree in nursing (F = 4.92, *p* < 0.01), work experience of over three years (F = 35.73, *p* < 0.01), or work experience in the current unit of over three years (F = 12.43, *p* < 0.01) had significantly higher levels of self-efficacy for handoffs (Table 1).

### 3.2. Levels of Self-Efficacy for Handoffs, Communication Competence, and IOCS

The mean values of self-efficacy for handoffs, communication competence, and IOCS were 36.41 (SD = 6.20), 53.45 (SD = 5.88), and 75.60 (SD = 11.63), respectively. In the four subdomains of IOCS, the highest mean was for the subdomain ‘vertical communication’ (Mean = 3.45, SD = 0.43), and the lowest was for ‘organisational climate’ (Mean = 2.54, SD = 0.57) (Table 2).

### 3.3. Correlations between Self-Efficacy for Handoffs, Communication Competence, and IOCS

Self-efficacy for handoffs was positively correlated with communication competence (r = 0.310, *p* < 0.001) and positively correlated with IOCS (r = 0.238, *p* < 0.001). In the four subdomains of IOCS, positive correlations between self-efficacy for handoffs and media quality and horizontal communication were significant (Table 3).

### 3.4. Effects of Communication Competence and IOCS on Self-Efficacy for Handoffs

This study identified the effects of communication competence and IOCS on self-efficacy for handoffs using stepwise multiple regression analysis. In the final regression model, the adjusted R square significantly explained 24.2% of the variance in self-efficacy for handoffs (F = 22.43, *p* < 0.001) when horizontal communication (*β* = 0.282, *p* < 0.001) from IOCS, work experience at the current unit (*β* = 0.215, *p* < 0.001), and communication competence (*β* = 0.180, *p* = 0.008) were included (Table 4). The regression model underwent an assessment to check for multicollinearity. The Durbin–Watson statistic registered at 1.948, nearly 2.0, suggesting an absence of autocorrelation in the residuals. Additionally, the variance inflation factors (VIF), ranging from 1.142 to 1.195 and well below 10, indicated that multicollinearity was not a concern in the model [30].

## 4. Discussion

In this study, satisfaction with horizontal communication was the most significant influencing self-efficacy for handoffs. It was challenging to compare this finding with previous results since those studies examined the effects of organisational communication on handoffs using a different research design, like qualitative research [17,19], or employed different instruments to assess organisational communication [18,31]. Nevertheless, some findings were comparable and consistent with ours, indicating that the more positive the relationship between incoming and outgoing nurses, the better the quality of handoff perceived by nurses [32,33]. Furthermore, in the nursing environment, which is dominated by a rigid and hierarchical atmosphere in South Korea, peer-to-peer communication satisfaction can help nurses feel psychological support and reduce handoff anxiety [34,35]. Our findings are aligned with other studies in cultural contexts similar to that of hospitals in South Korea; several novice nurses in Hong Kong felt more nervous when reporting to senior nurses, who were more experienced and often experienced difficulties communicating with their seniors during the handoffs [17]. In Saudi critical care settings, 40 out of 151 participants reported conflicts and poor relationships among nurses; in addition, they perceived the good relationship between nurses as the most significant predictor of handover quality, alongside cognitive capacity and focused attention [31].

This finding can be explained by other studies reporting that, as one of the main organisational climates, positive horizontal communication can reduce risk since incoming nurses are more likely to ask questions and clarify information with outgoing nurses, which can reduce information omissions and misinterpretation [4,18]. Pun’s [4] study on factors associated with the quality of nursing handoffs found that opportunities to ask questions significantly impacted their understanding of patient care plans, facilitating nurses’ perception of the quality of handoffs. Thomson et al. [32] also suggested that incoming nurses are more likely to feel comfortable asking questions and clarifying information with their outgoing colleagues through the development of positive relationships with colleagues. Moreover, nurses who report higher levels of satisfaction with communication can have higher levels of self-efficacy for handoffs as a result of successful handoff experiences based on Bandura’s self-efficacy theory [36,37]. Given this, such empirical evidence implies that the quality of handoffs depends not only on the transmission of technical facts but also on all interpersonal behaviours that help create a mutually cooperative mood and an effective dialogue between nurses. Organisations and nurse managers should be responsible for creating a positive organisational atmosphere; they need to pay attention to a positive organisational climate regarding the relationships among nurses so that nurses can communicate with their peer nurses more comfortably and trustfully.

As for another significant finding, it is noteworthy that nurses with better communication competence had higher self-efficacy for handoffs. This supports previous studies showing that communication skills significantly influence nurses’ perceptions of the quality of handoffs [4,38]. This consistency in findings can be explained by Cegala’s communication competence concepts [39]. Good communication skills in information verification include clarifying, repeating, summarising, and forecasting information that may be given or asked for later [39]. These behaviours are essential during a handoff to reduce the chances of misunderstanding by ensuring comprehension [15]. A study on analysis of communication behaviours associated with the competent nursing handoff among 286 American nurses by Streeter et al. [15] revealed that competent nursing handoffs were highly associated with good communication skills, such as information exchange and socioemotional communication behaviours.

Furthermore, Eggins and Slade [40] indicated that the quality of handoffs relies on nurses’ communication competence rather than the application of a standardised handoff tool. Such consistent findings can be supported by the concept of locus of control in self-efficacy [41,42]. That is, nurses’ communication competence can be considered an internal locus of control as one of the personal factors influencing self-efficacy. The internal locus of control pertains to an individual’s beliefs that outcomes are largely the result of their actions and decisions [42]. Individual nurses with a strong internal locus of control often have higher self-efficacy. They believe their actions can cause desired outcomes, so they are more likely to feel confident in their abilities to tackle challenges and achieve their goals [41].

Nonetheless, environmental factors influencing self-efficacy for handoffs, such as the physical environment related to handoffs or standardised guidelines and processes, should also be contemplated. These factors can be a strong external locus of control, decreasing handoff self-efficacy. External locus of control refers to individual beliefs about outcomes resulting from external factors such as other people’s actions or self-uncontrolled environmental causes [41]. According to a recent integrated review of nurse-to-nurse handoffs, the most common types of interruptions and distractions were people (patient or family) and environmental factors (equipment alarm) [1]. Raeisi et al. [33] recommended the application of standardised protocols and systemic approaches to handoffs based on unit and organisational needs in their systematic review. Oh and Gastmans [43] pointed out that nurses experienced difficulties in caring for patients due to a lack of information about the novel disease or inconsistent healthcare policies during the COVID-19 pandemic. Such environmental factors can influence nurses’ self-efficacy for handoffs; thus, studies are suggested to compare this self-efficacy before and after a public health crisis like a pandemic. Further research should include personal, environmental, and structural factors as influencing factors and examine their impact on self-efficacy for handoffs among nurses in various care settings.

The last significant factor influencing handoff self-efficacy was years of work experience in the current unit. This finding is aligned with previous studies on nurses’ perceptions of the quality of handoffs. Rhudy et al. [19] reported that nurses with more work experience can readily identify peers with different approaches and preferences than their own. Abou Hashish et al. [31] found that nurses’ experience in documentation was one of the major determinant factors of handoff quality despite using the same standardised tool. Conversely, some studies reported no significant impact of nurses’ work experience on the quality of handoffs [4,32]. This contrast might be caused by using handoff tools, supporting systems, or programmes of varying quality to develop handoff reporting competency. Further studies are needed to investigate the association between these variables in various care settings or countries.

## 5. Implications and Limitations

From the perspectives of personal and environmental factors influencing self-efficacy, our study offers compelling empirical evidence to support and confirm the self-efficacy theory. This is achieved by verifying the effect of communication competence and satisfaction derived from intra-organisational communication on nurses’ self-efficacy during handoffs. Moreover, our findings provide instrumental insights into the necessary components for crafting education programmes geared toward proficient nursing handoffs. Despite the strengths and implications of our study, certain limitations are present. First, convenience sampling may curtail the generalisability of our results, even though we gathered data from five different hospitals spanning diverse regions. Second, most participants were female since male Korean nurses accounted for only 5.1% of registered nurses in 2019 [44]. This suggests potential gaps in capturing a balanced representation of gender dynamics and disparities. Further research is required to illuminate communication competence and satisfaction with organisational communication related to self-efficacy for handoffs among male nurses and achieve gender equity based on standards such as the Sex and Gender Equity in Research guidelines [45]. Third, 24.2% of the R square value could be limited in interpreting our model; however, this could be acceptable and sufficient to interpret the psychological experience among nurses since most social science research modelling aims not to predict human behaviours [46]. Instead, the goal is often to assess whether specific predictors or explanatory variables significantly affect the dependent variable [47]. Finally, data were obtained using self-report questionnaires. However, this concern could be allayed by employing psychometrically sound measures [48].

## 6. Conclusions

This study revealed the significantly influential self-efficacy factors for handoffs perceived by nurses. In self-efficacy for handoffs, as perceived by nurses, satisfaction with communication among nurse colleagues, and an individual’s communication competence are vital. In addition, work experience in the current unit is included. To improve self-efficacy for nurse-to-nurse handoffs, organisations and nurse managers should foster a positive organisational climate for relationships among nurses so that nurses feel satisfied communicating with peer nurses. They should also provide opportunities, such as education programmes or systems, for nurses to develop their communication competence. Moreover, it is suggested that nurses with more work experience in the current unit can be role models to novice nurses for nurse-to-nurse handoffs.

## Figures and Tables

**Table 1 healthcare-11-03125-t001:** Differences in self-efficacy for handoffs based on general characteristics (*n* = 203).

Characteristics	Categories	*n* (%)		Mean (SD)		Self-Efficacy for Handoff			
						Mean	(SD)	t or F	*p*
								Post Hoc Scheffé Tests	
Gender	Female	198	(97.5)			3.69	(0.68)	−0.28	0.78
	Male	5	(24.6)			3.78	(1.06)		
Age (year)	20–25 ^a^	69	(34.0)	28.5	(5.6)	3.33	(0.67)	13.37	<0.01
	26–30 ^b^	92	(45.3)			3.73	(0.56)	a, < b, c, d, e	
	31–35 ^c^	18	(8.9)			4.20	(0.72)		
	36–40 ^d^	12	(5.9)			4.19	(0.69)		
	40 ^e^ and older	12	(5.9)			4.22	(0.44)		
Marital status	Unmarried	166	(82.6)			3.60	(0.69)	−5.10	<0.01
	Married	35	(17.4)			4.13	(0.53)		
Religion	With a religion	93	(45.8)			3.67	(0.15)	0.84	0.47
	Without a religion	110	(54.2)			3.72	(0.73)		
Educational level	Associate degree ^a^	50	(24.7)			3.65	(0.79)	4.92	<0.01
	Bachelor’s degree ^b^	141	(69.5)			3.66	(0.65)	a, b < c	
	Graduate school ^c^	12	(5.9)			4.29	(0.48)		
Workplace	Internal medicine ward	89	(43.8)			3.69	(0.72)	0.78	0.46
	Surgical medicine ward	69	(34.0)			3.63	(0.71)		
	General medicine ward	45	(22.2)			3.80	(0.60)		
Nursing experience (year)	<1 ^a^	49	(24.1)	4.8	(5.7)	3.10	(0.65)	35.73	<0.01
	1 to 3 ^b^	60	(29.6)			3.57	(0.57)	a < b < c, d	
	3 to 6 ^c^	44	(21.7)			3.92	(0.50)		
	≧6 ^d^	50	(24.6)			4.21	(0.51)		
Work experience in the current unit (year)	<1 ^a^	74	(36.5)	2.3	(1.9)	3.37	(0.75)	12.43	<0.01
	1 to 3 ^b^	79	(38.9)			3.76	(0.60)	a < b < c, d	
	3 to 6 ^c^	37	(18.2)			4.04	(0.47)		
	≧6 ^d^	13	(6.4)			4.15	(0.56)		
Time for preparation for a handoff (min)	<30	78	38.4			3.86	(0.28)	2.84	0.01
	≧30	125	61.6			3.59	(0.63)		
Length of time for a handoff (min)	<30	87	42.9			3.83	(0.59)	2.47	0.01
	≧30	116	57.1			3.59	(0.74)		

a, b, c, d, and e: groups from post hoc Scheffé test.

**Table 2 healthcare-11-03125-t002:** Levels of self-efficacy for handoffs, communication competence, and IOCS (*n* = 203).

Variables	Mean (SD)	Range	Min–Max
Self-efficacy for handoffs	36.41 (6.20)	9–54	10.00–54.00
Communication competence	53.45 (5.88)	15–75	39.00–70.00
IOCS	75.60 (11.63)	0–336	50.00–103.00
Vertical communication	3.45 (0.43)	1–5	2.25–4.50
Horizontal communication	3.00 (0.54)	1–5	1.40–4.40
Media quality	3.09 (0.53)	1–5	1.50–5.00
Organisational climate	2.54 (0.57)	1–5	1.00–4.00

**Table 3 healthcare-11-03125-t003:** Correlation between self-efficacy for handoffs, communication competence, and IOCS (*n* = 203).

	Self-Efficacy for Handoffs	
	r	*p*
Communication competence	0.310	<0.001
IOCS	0.238	0.001
Media quality	0.248	<0.001
Horizontal communication	0.242	<0.001
Vertical communication	0.138	0.050
Organisational climate	0.080	0.260

**Table 4 healthcare-11-03125-t004:** Effects of communication competence and IOCS on self-efficacy for handoffs (*n* = 203).

Independent Variables	Unstandardised Coefficient		Standardised Coefficients			Collinearity Statistics	
	B	S.E.	*β*	t	*p*	Tolerance	VIF
(constant)	14.882	3.917					
IOCS							
Horizontal communication	2.783	0.699	0.282	3.980	<0.001	0.875	1.142
Work experience in the current unit	0.048	0.015	0.215	3.270	<0.001	0.750	1.333
Communication competence	0.223	0.083	0.180	2.681	0.008	0.837	1.195

F = 22.43, *p* < 0.001, *R*^2^ = 0.254, Adjusted *R*^2^ = 0.242, *d*(*d_u_*) = 1.948.

## Data Availability

Data can be made available upon request from the corresponding author of this study.

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
