# Peer review of "Effects of Communication Skills and Organisational Communication Satisfaction on Self-Efficacy for Handoffs among Nurses in South Korea"

_healthcare, 2023, doi:10.3390/healthcare11243125_

Round 1

Reviewer 1 Report

Comments and Suggestions for Authors

General issues:

The aim of the study is to identify the association between self-efficacy for handoffs and communication competence and between self-efficacy for handoffs and intra-organizational communication satisfaction and the impact of competence and intra-organizational communication satisfaction on self-efficacy for handoffs among nurses [lines : 96-99]. Cross-selectional analysis was used [line: 102]. The study was conducted on a sample of 206 full-time registered nurses from five general hospitals with more than 200 beds in the three major regions of South Korea: Seoul, Gyeonggi, and Gang won Province [lines: 102-105]. The study was conducted between September 20 and October 9, 2018 [line: 111]. A high survey response rate of 99% was achieved [lines: 115-116]. The Korean version of the Perceived Self-efficacy of Handoff reporting (K-PSH) [lines: 127-128] and a Global Interpersonal Communication Competence (GICC) [lines: 140-141] and a Korean version of the Intra-Organizational were also used Communication Satisfaction (K-IOCS) [lines: 148-149]. Statistical methods were used, i.e. Cronbach's alpha reliability analysis, t-test, one-way analysis of variance (ANOVA), with post hoc Scheffé tests, Pearson's correlation coefficients, and multiple regression analysis [lines: 157-159].

Detailed issues:

Lines 112-115: Two sentences refer to the same issue, i.e. how to collect data. One of the sentences should be deleted, e.g. 'The questionnaires were collected by the researcher during hospital visits or by mail'.

Line 160: Incorrect subsection number - it should be 2.4.

Lines 168-169: Maybe it would be worth considering changing the title of subsection 3.1. Maybe: 'General characteristics of the sample(…)'

Lines: 184-185: In Table 1, n(%) is indicated in the third column, which would suggest that the values are expressed in percentages, while these are numbers for individual features. It may be worth providing a separate description for the third and fourth columns. It would also be worth clearly marking in the table where the average is indicated and what it concerns. In the case of age 28.5, it is unsigned and would indicate that the average size of the group is 28.5 people and not that the average age is that [similarly in: Nursing experience, Work experience in the current unit]. You should also consider changing the name of the table, as well as the title of the subsection.

 Summary:

This is a very interesting study, but it raises some doubts. It should be noted that the study was conducted in 2018, i.e. 5 years ago. In connection with the above, it should be considered whether these results do not differ from reality, in particular due to the period of the Covid-19 virus pandemic, which could have had a significant impact on the behavior of medical staff, in particular nurses. It is worth mentioning very low correlations among self-efficacy for handoffs, communication competence, and IOCS, the highest of them is only 0.310 [lines: 198-199]. The study also achieved a very low R squared level of only 24.2%, which indicates poor model fit [lines: 202-206]. Despite this, very categorical conclusions are made regarding the relationships between variables. It seems that it would be worth expanding the chapter on the conclusions of the study, as it is very general. It would be worth including a sample survey questionnaire as an attachment to the article.

Author Response

We thank the reviewer very much for the constructive criticism that helped us improve the quality of our manuscript. We have revised the manuscript in accordance with the reviewer’s comments. All changes made in the manuscript are highlighted in yellow.

Reviewer 2 Report

Comments and Suggestions for Authors

Dear Authors,

The work is well organised, results are presented in detail using tables and a limitation paragraph is included at the end of article. Those parameters create a very nice flow.

My main concern is the year of study back in 2018. On the experimental section it mentions clearly that 'The 206 participants were assessed between 20 September and 9 October 2018, and 111 the questionnaires were distributed to the nurses'.

The results need an update and probably the addition of fresh data that can be combined with the existing. So, I suggest to share the questionnaire again, obtain data and compare old and new data. Probably, a comparison prior and after Covid-19. In the current, format I  believe its not publishable due to the late date back in 2018.

Also the first paragraph mentions the pandemic, however the study was conducted prior to this global crisis. Again the pandemic should be addressed in more detail on the intro as well.

Author Response

We thank the reviewer very much for the suggestions that helped us improve the quality of our manuscript. We have revised the manuscript in accordance with the reviewer’s comments. All changes made in the manuscript are highlighted in yellow.

Round 2

Reviewer 1 Report

Comments and Suggestions for Authors

The authors have corrected or addressed issues identified in the first review. The article may be published in its current form.

Reviewer 2 Report

Comments and Suggestions for Authors

Dear authors,

The comments have been addressed and all the necessary clarifications have been made. Thus, I am recommending acceptance.